# Aligning restricted access data with FAIR: a systematic review

Margherita Martorana, Tobias Kuhn, Ronald Siebes and
Jacco van Ossenbruggen

Department of Computer Science, Vrije Universiteit Amsterdam, Amsterdam, Netherlands

## ABSTRACT

Understanding the complexity of restricted research data is vitally important in the current new era of Open Science. While the FAIR Guiding Principles have been introduced to help researchers to make data Findable, Accessible, Interoperable and Reusable, it is still unclear how the notions of FAIR and Openness can be applied in the context of restricted data. Many methods have been proposed in support of the implementation of the principles, but there is yet no consensus among the scientific community as to the suitable mechanisms of making restricted data FAIR. We present here a systematic literature review to identify the methods applied by scientists when researching restricted data in a FAIR-compliant manner in the context of the FAIR principles. Through the employment of a descriptive and iterative study design, we aim to answer the following three questions: (1) What methods have been proposed to apply the FAIR principles to restricted data?, (2) How can the relevant aspects of the methods proposed be categorized?, (3) What is the maturity of the methods proposed in applying the FAIR principles to restricted data?. After analysis of the 40 included publications, we noticed that the methods found, reflect the stages of the Data Life Cycle, and can be divided into the following Classes: Data Collection, Metadata Representation, Data Processing, Anonymization, Data Publication, Data Usage and Post Data Usage. We observed that a large number of publications used 'Access Control' and 'Usage and License Terms' methods, while others such as 'Embargo on Data Release' and the use of 'Synthetic Data' were used in fewer instances. In conclusion, we are presenting the first extensive literature review on the methods applied to confidential data in the context of FAIR, providing a comprehensive conceptual framework for future research on restricted access data.

## INTRODUCTION

In the last 10 years, the role of Open Science in scientific research has received considerable attention across several disciplines, and a growing body of literature has been proposed to implement Openness. Evidence suggests that the replication of results, the discovery and exchange of information, and the reuse of research data have emerged as some of the most important reasons for Open Science (*Hey et al., 2009*; *Wilkinson et al., 2019*). Interestingly, the reuse of "data created by others", also known as secondary data, described as "the basis of scholarly knowledge" (*Pampel & Dallmeier-Tiessen, 2014*), is



Corresponding author
Margherita Martorana,
m.martorana@vu.nl

considered one of the key aspects of Open Science (*Vicente-Sáez & Martnez-Fuentes, 2018*).

To facilitate the creation, management and usage of secondary data, several initiatives have been involved in building solutions for Open Science research. For instance, the Center for Open Science (https://www.cos.io) has proposed the Open Science Framework (OSF) (*Foster & Deardorff, 2017*), to promote an open tool for the storage, development and usage of secondary data. The LIGO Open Science Center (LOSC) (https://losc.ligo.org) is another initiative, with the intent to facilitate Open Science research in the Astronomy domain by offering a platform to discover and analyse data from the Laser Interferometer Gravitational-wave Observatory (LIGO) (*Vallisneri et al., 2015*). Archives such as GenBank (*Benson et al., 2012*) and *menoci* (*Suhr et al., 2020*) have also made their data open and accessible to users in the biomedical domain, and EUDAT (*Widmann & Thiemann, 2016*) is an international initiative to help overcome the challenges related to the reuse of data, by offering a Collaborative Data Infrastructure (CDI) for the research community. Other developments, in the effort to render the technical difficulties linked to the use of secondary data, have been in 2016 with the introduction of the FAIR Guiding Principles (*Wilkinson et al., 2016*). The Principles aim to provide guidelines for making data Findable, Accessible, Interoperable and Reusable. The implementation of FAIR has been demonstrated to improve data management and stewardship (*Boeckhout, Zielhuis & Bredenoord, 2018*; *Mons, 2018*), by enabling the reuse of data, promoting collaborations and facilitating resource citation (*Lamprecht et al., 2020*). Ensuring transparency, reproducibility and reusability can also help data owners and publishers to define data sharing plans and to improve the discoverability of resources (*Wilkinson et al., 2017*).

Several studies have shown how to implement Open and FAIR data, but not all data is the same and not all data is suitable for being publicly available. For example, medical records and patients' data contain, by nature, Personal Identifiable Information (PII) if not sanitized. Government data, such as census data and other types of information retrieved by governmental agencies about the population, are often not open to the public because of confidential concerns. Despite recent attempts from the European Union to provide methods for dealing with personal and confidential information, *i.e.*, GDPR, there are still considerable limitations that have not yet been fully investigated. Regulations can often be vague, ambiguous and not well defined. For instance, the GDPR requires data owners and stakeholders to provide a 'reasonable level of protection', without clearly specifying what the word 'reasonable' actually involves. Also, the concept of 'privacy by design' is consistently supported within the regulations, but no clear guidelines on how to achieve it are proposed. Overall, legal compliance in the context of restricted and privacy concerning data is most often a challenge, first by determining what are the regulations to comply with and second by having the technical ability to guarantee such compliance (*Otto & Antón, 2007*). To date, it remains unclear how sensitive data should be managed, accessed and analysed (*Cox, Pinfield & Smith, 2016*; *Leonelli et al., 2021*; *Bender, Blaschke & Hirsch, 2022*).

Researchers have claimed that data containing confidential and private information should not be made open, and its access should be tightly regulated (*Levin & Leonelli, 2017*). This notion comes from the fact that data has been seen, so far, to have a binary state of either open or closed. The FAIR Principles, on the other hand, do not provide an all-or-nothing view on the data (either FAIR or not FAIR), but they represent more of a guideline and a continuum between data being less FAIR and more FAIR (*Betancort Cabrera et al., 2020*). Moreover, FAIR data, and more specifically Accessible data, does not necessarily require to be also open (*Mons et al., 2017*). Accessible data can be defined as such when once fulfilled certain requirements, the data can be made either partially or fully accessible. More in detail, data access can be mediated through automated authorization protocols (*Wilkinson et al., 2016*) as well as through direct contact with the data owner, but as long as access to the data can, in theory, be achieved, then that data can be considered as accessible (*Gregory et al., 2019*). As mentioned above, the Principles do not define a binary state of either FAIR or non-FAIR data, between accessible and inaccessible, open and closed. Instead they define guidelines for the "optimal choices to be made for data management and tool development" (*Betancort Cabrera et al., 2020*). The application of the FAIR Guiding Principles to government and confidential data does not have the aim to make it publicly open but, indeed, to make it more Findable, Accessible, Interoperable and Reusable. At this point, it is important to clearly define what the authors of this paper mean when referring to the term 'restricted access data'. In the context of this paper, such a term will be abbreviated to 'restricted data' and refers to any type of datasets, artefacts or digital objects which is not freely available (*e.g.*, medical records, patient data and government data). The lack of accessibility can either be determined by confidential and privacy protection regulations, as well as usage and license terms.

While the Principles have sparked many international and interdisciplinary initiatives, such as GO FAIR (https://www.go-fair.org), the Commission on Data of the International Science Council (CODATA) (https://codata.org) and the Research Data Alliance (RDA) (https://www.rd-alliance.org), most of the application of FAIR have been seen in the "hard" sciences (*e.g.*, biology, astronomy, physics). There is still a lack of understanding and specific recommendations on how FAIR can be implemented more in Social, Behavioural and Economic (SBE) domains, also referred to as "soft" sciences. SBE scientists are often faced with many domain-specific challenges, often linked to the various data collection methods and data types used. In fact, SBE sciences generally require data from questionnaires, interviews or surveys which are usually gathered from public institutions such as official registries and government bodies. Therefore, it is highly likely that the data contain personal and confidential information that can disclose the identity of individuals and institutions. Before the data can be used for analysis and shared with researchers, the data owner is responsible for assuring that the confidentiality of the data subjects is kept intact and is not at risk, and this process is usually performed by anonymizing the data and implementing strict access control policies. Nevertheless, this process is often not completely transparent and can require strenuous bureaucratic steps for the researchers before gaining access. Moreover, there is still no consensus within the

scientific community about what are the methods and procedures recommended when dealing with restricted data. The purpose of this investigation is to explore the relationship between FAIR and restricted data and to assess the mechanisms for making restricted data (more) FAIR, to facilitate the reuse and discoverability of secondary data.

## Problem statement and contributions

The primary aim of this review is to investigate the methods employed by data owners and users when dealing with restricted research data, in the context of the FAIR principles. Understanding the complexity of reusing restricted data is crucial in a variety of fields, from the biomedical to the social science domain.

The present review provides the first comprehensive assessment of the relationship between the FAIR Guiding Principles and restricted data, by answering the following questions:

*What methods have been proposed to apply the FAIR principles to restricted data?*
*How can the relevant aspects of the methods proposed be categorised?*
*What is the maturity of the methods proposed in applying the FAIR principles to restricted data?*

This work contributes to the existing knowledge of FAIR by providing an extensive framework describing the methods employed when researching restricted data. With this review, we are laying the groundwork for future research into making restricted data more Findable, Accessible, Interoperable and Reusable. Moreover, the categorisation of methods and the ontology created based on the results can provide a reusable framework to express the methods used when dealing with restricted data.

The remaining part of the paper proceeds as follows: the next section begins by illustrating the background information about the role of Data Science, Restricted Research Data and the FAIR Guiding Principles within the scope of this review. We will then describe the methods used for the selection and analysis of the included articles and the following sections will present and discuss the results. The last section of the review will summarise the overall findings and provide a final overview of the relationship between FAIR and restricted data.

## BACKGROUND

### Data science

Data Science is fast becoming a key component in nearly all scientific domains and industrial processes, and in the last decade, it has emerged as a research field of its own. As more data is produced, and more analysis techniques are made available, there is a need for specialised skills to embark on this data-dependent world. Industries, as well as scientific fields such as Medicine and Engineering, have now become data-driven and their success is closely related to their ability to explore and analyse complex data sources (*National Academies of Sciences, Engineering & Medicine, 2018*). The application of Data Science has already been seen in known data-intensive fields, for example, Geoscience (*Singleton & Arribas-Bel, 2021*), Biology (*Shi, Aihara & Chen, 2021*) and Artificial

Intelligence (*Sarker et al., 2021*). Nevertheless, less data-driven domains are also adapting to the Data Science wave, creating new fields of studies such as Digital Humanities and Computational Social Sciences.

## Restricted access data

Recently, we have increasingly seen the development of online Open Government Data (OGD) portals, intending to enhance innovation, economic progress and social welfare. Through the creation of OGD, governments have allowed the general public to easily access information that was long thought unattainable, and use them in a variety of fields such as journalism, software development and research (*Begany, Martin & Yuan, 2021*). The economic value of public records has been expected, by the European Commission, to increase from 52 billion in 2018 to 215 billion in 2028 (*Barbero et al., 2018*), thus emphasizing the economic impact on different public sectors, such as transportation, environmental protection, education and health (*Quarati, 2021*).

Although the beneficial contribution of using public records to the overall well-being of society is clear, there is still a large amount of data that can not be made publicly available due to confidentiality concerns. For example, health data from hospitals and medical centres are often restricted to the data owners and stakeholders, due to patient privacy concerning issues. Moreover, researchers can also decide not to make their data open, and instead, apply limitations to their use through usage terms and licenses.

## FAIR guiding principles

Since the publication of FAIR in 2016, there is a growing number of literature that applies the Guiding Principles and recognises the importance of making data Findable, Accessible, Interoperable and Reusable. The FAIR initiative is taking up more and more momentum, and the application of the Principles has been seen in nearly all fields of science. For example, a recent paper by *Kinkade & Shepherd (2021)* proposes practical solutions to address and achieve FAIR in data-driven research in geosciences. Another recent paper discusses the importance of the Principles in expanding epidemiological research within the veterinary medicine domain (*Meyer et al., 2021*). The FAIR guidelines have also been applied in more technical fields such as the scientific research software domain. A community effort to improve the sharing of research software has brought the creation of the FAIR for Research Software (FAIR4RS) Working Group (*Chue Hong et al., 2021*; *Katz, Gruenpeter & Honeyman, 2021*) and the "Top 10 FAIR Data & Software Things" (*Erdmann et al., 2019*). Other examples of guidelines that have stemmed from the original FAIR Principles are the "FAIR Metrics" (*Wilkinson et al., 2018*) and the "FAIR Data Maturity Model" (*FAIR Data Maturity Model Working Group, 2020*).

The FAIR Guiding Principles have also been gaining increasing interest and recognition from international entities such as the European Commission and the National Institute of Health (NIH) (https://www.nih.gov). The latter, together with the Department of Health and Human Services (HHS) (https://www.hhs.gov) and the Big Data to Knowledge (BD2K) initiative (*Margolis et al., 2014*), are supporting the application of FAIR in the biomedical domain through the development of innovative approaches to big data and

data science. The European Commission has particularly been involved in the application of the Principles through international initiatives, such as the Internet of FAIR Data and Services (IFDS) (https://www.go-fair.org/resources/internet-fair-data-services/) and the European Open Science Cloud (EOSC) (https://eosc-portal.eu), to implement strategies for the application of FAIR on digital objects, technological protocols, digital data-driven science and the Internet of Things (*European Commission, 2016*; *van Reisen et al., 2020*). Moreover, the European Commission is now mandating the use of FAIR in new projects, and it is working towards comprehensive reports and action plans for 'turning FAIR data into reality' (*European Commission, 2018*).

Nevertheless, the technical implementation of FAIR is still the main challenge faced by many stakeholders. The FAIR principles call for data to be both machine and human-readable to facilitate the retrieval and analysis of resources. This process requires the data stakeholder to generate a machine-readable format of the data, often using the Resource Description Framework (RDF) (*Miller, 1998*). Further, another core principle of FAIR is the importance of not only data standards but also metadata standards. The term 'metadata' refers to the top-level information and attributes describing the data, such as the provenance, the methodology used as well as terms of use of the artefact. More in general, metadata can be thought of as the bibliographic information about the data is describing (*Boeckhout, Zielhuis & Bredenoord, 2018*). Yet, the process by which FAIR metadata should be generated and organised to include all relevant information is still unclear. On top of the need for clear technical guidelines for the implementation of the Principles, there is also the need of changing the work culture to mirror the core meaning of FAIR. From a business point of view, enterprises need evidence to show how they can generate a long-term return on investment (RoI) through the application of FAIR, and for research centres, management boards are still to be convinced about the benefits brought by the Principles, such as peer-recognition, data accessibility and financial rewards (*Wise et al., 2019*; *Stall et al., 2019*).

It is clear how the impact of the FAIR Principles is important for the future of Open Science, technological development and scientific research. With the continuously growing number of domains where the Principles are applied and the increasing amount of data generated, it is essential to understand the mechanisms of FAIR in the context of restricted data. In this review, we aim to provide a better understanding of how the FAIRification process can benefit restricted data, by analysing the methods employed by the scientific community to overcome the barriers of confidentiality and to guide research on privacy concerning data toward mature FAIR choices.

## METHODS

In the following section, we describe the methods employed in the review, by first describing details of the resources' selection step and the application of inclusion and exclusion criteria. We then provide information concerning the data collection and data analysis processes. We aim to investigate the common methods utilised to overcome issues related to restricted data, in the context of FAIR-driven research.

## Eligibility criteria

Several criteria were considered when selecting the studies to be included in the analysis. Eligibility criteria required articles to:

1. be written in English.
2. be peer-reviewed.
3. be research papers.
4. clearly describe the proposal or application of the FAIR principles in the context of restricted data.

The selection of the eligibility criteria was made based on a few conditions. English is the lingua franca of science communication and was the only language shared by all authors, and therefore only papers written in English were included in the systematic review. Secondly, we decided to exclude papers that were not peer-reviewed or other systematic reviewed, therefore only including peer-reviewed research papers. The last eligibility criterion was formulated based on the fact that the authors wanted to include papers that showed a clear application of FAIR principles concerning restricted data, rather than just the mentioning of such without an apparent use of the principles.

## Search strategy

The electronic literature search for this study was conducted on the Google Scholar database on the 16th of September 2021, using the following query:

"findable accessible interoperable reusable" AND (copyrighted OR confidential OR sensitive OR restricted OR privacy)

No other filters or limits were set in the search.

## Selection process

The *corpus* of publications resulting from the query was exported to Rayyan (*Ouzzani et al., 2016*), a Software as a Service (SaaS) web application used for the screening of publication data. The tool was used solely for the management of publications and to help resolve duplicates. The application of inclusion and exclusion criteria, as well as the resolution of duplicated publications, is not dependent on Rayyan, and the same results are to be expected if another citation management tool is used. The first author of this paper (M. Martorana) independently screened each record to evaluate their eligibility. The process of evaluation started with reading the abstract of each publication. If a decision on its eligibility could not be made, the whole paper was read.

## Data collection and analyses

A qualitative approach was adopted to capture descriptive data from the included publications.

The first step in this process was to determine the Field of Research each paper belonged to. Second, information regarding the suggestion or application of the methods was collected. On completion, an iterative process was carried out to group the outcomes into

**Table 1** The table shows the grouping of the technology readiness levels (TRLs) based on the work by (*European Commission, 2017*), and it provides a definition on how each TRL group was assigned to theincluded publications.

| Technology readiness levels based on (*European Commission, 2017*) | Definition |
| --- | --- |
| TRL 1 & 2 | This class was assigned to publications where the technology proposed was only conceptually formulated but not implemented. |
| TRL 3 & 4 | This class, instead, was assigned to publications where the technology proposed has gone through some testing but only in limited environments. |
| TRL 4 & 5 | This class was assigned to publications that clearly showed testing and expected performance. |
| TRL 7, 8 & 9 | Lastly, this class was assigned to publications that showed full technical capabilities and that were also available to users. |

concrete Classes, which were then recognised to resemble different stages of the Data Life Cycle. By the term "Data Life Cycle", we refer to the stages the data goes through from the moment of collection to what happens after its usage. It is important here to distinguish between the Data Life Cycle steps recognised in this research, and the most common steps most often identified as the Data Lifecycle Management. The latter, represents an overview of the steps the data owners would mostly be faced with during the management and safekeeping of data, and they involve: (1) creating data, (2) data storage, (3) data use, (4) data archive and (5) data destruction. In the context of this research, the cycle has been aligned to the data lifecycle management, but we have also added steps to include the stages when the data is processed, as well as the steps required after the data is used. Next, each publication was annotated concerning the methods proposed. Finally, a technology readiness level (TRL) (*Héder, 2017*), based on the maturity of the technology proposed, was estimated for each publication. The appraisal of TRLs offers an effective way to assess how different technical solutions related to more advanced research infrastructures, by assigning a score representing the level of maturity of the technology. For practical purposes, as well as to decrease potential miscalculation of the scores, the TRL levels were organised into the following four main groups based on the European Union definitions (*European Commission, 2017*). Table 1, below, shows how the TRL levels were grouped, and define the level of maturity expressed by each group.

## Synthesis

The final stage of the methodology comprised of a visual representation of the publications and their relative methods and TRL scores, as well as the creation of an OWL ontology representing the methods. The Web Ontology Language (OWL) (*Antoniou & van Harmelen, 2004*) was used to provide a FAIR representation of the results of this systematic review in the form of a human and machine readable "Data Methods" ontology. The ontology we created could be used in the future to help describe the methods implied when researching restricted data in a FAIR manner. The decision of building our ontology in OWL is based on the fact that it is a W3C approved semantic language, designed to formally define rich meaning and concepts.

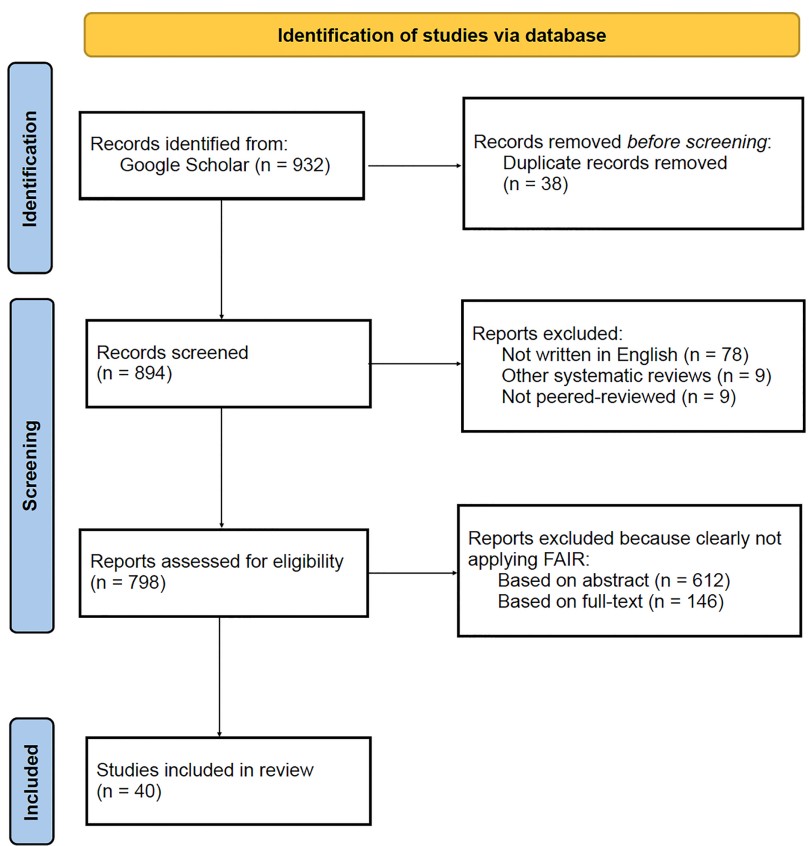

**Figure 1 PRISMA flow diagram illustrating the results from the search and selection process, performed on the Google Scholar database.**

# RESULTS

## Study selection

The first set of results concerns the outcome of the search and selection process of papers. Google Scholar returned an overall number of 932 publications based on the query performed. Duplicates were detected and resolved accordingly, resulting in 894 unique publications. Further, publications were excluded based on the following criteria: 78 were not written in English, nine were not peered-review and the other nine were systematic reviews. The remaining 798 papers were screened, and 612 were excluded based on the abstract, and 146 based on the full text. Ultimately, 40 publications were included in the review for data extraction and analysis. A summary of these results can be found below, in Fig. 1. Details of the 40 publications included in the review can be found below, in Table 2.

## Field of research

The first set of analyses examined the 'Field of Research' each of the included publications belonged to. We were able to distinguish nine different fields, and we found that the 'Biomedical Domain' was the most common field of research, with 27 papers (67.5%) linked to it. We also found that 5% of papers were linked to the 'Biodiversity' domain, and

**Table 2 List with short authors, year and title references of the final 40 publications included in the systematic review.**

| Authors | Year | Title |
| --- | --- | --- |
| Dyke et al. | 2016 | Consent codes: upholding standard data use conditions |
| Lakerveld et al. | 2017 | Identifying and sharing data for secondary data analysis of physical activity, sedentary behaviour and their determinants across the life course in Europe: general principles and an example from DEDIPAC |
| Bertocco et al. | 2018 | Cloud access to interoperable IVOA-compliant VOSpace storage |
| Kleemola et al. | 2019 | A FAIR guide for data providers to maximise sharing of human genomic data |
| Sun et al. | 2019 | A Privacy-Preserving Infrastructure for Analyzing Personal Health Data in a Vertically Partitioned Scenario. |
| Rockhold et al. | 2019 | Open science: The open clinical trials data journey |
| Demotes-Mainard et al. | 2019 | How the new European data protection regulation affects clinical research and recommendations? |
| Van Atteveldt et al. | 2019 | Computational communication science\| toward open computational communication science: A practical road map for reusable data and code |
| Dimper et al. | 2019 | ESRF Data Policy, Storage, and Services |
| Lahti et al. | 2019 | 'As Open as Possible, as Closed as Necessary'-Managing legal and owner-defined restrictions to openness of biodiversity data. |
| Becker et al. | 2019 | DAISY: A Data Information System for accountability under the General Data Protection Regulation |
| Kephalopoulos et al. | 2020 | Indoor air monitoring: sharing and accessing data *via* the Information Platform for chemical monitoring (IPCHEM) |
| Hoffmann et al. | 2020 | Guiding principles for the use of knowledge bases and real-world data in clinical decision support systems: report by an international expert workshop at Karolinska Institutet |
| Cullinan et al. | 2020 | Unlocking the potential of patient data through responsible sharing–has anyone seen my keys? |
| Nicholson et al. | 2020 | Interoperability of population-based patient registries |
| Paprica et al. | 2020 | Essential requirements for establishing and operating data trusts: practical guidance co-developed by representatives from fifteen Canadian organizations and initiatives |
| Jaddoe et al. | 2020 | The LifeCycle Project-EU Child Cohort Network: a federated analysis infrastructure and harmonized data of more than 250,000 children and parents |
| Bader et al. | 2020 | The International Data Spaces Information Model–An Ontology for Sovereign Exchange of Digital Content |
| Aarestrup et al. | 2020 | Towards a European health research and innovation cloud (HRIC) |
| Suver et al. | 2020 | Bringing Code to Data: Do Not Forget Governance |
| Roche et al. | 2020 | Open government data and environmental science: a federal Canadian perspective |
| Beyan et al. | 2020 | Distributed analytics on sensitive medical data: The Personal Health Train |
| Choudhury et al. | 2020 | Personal health train on FHIR: A privacy preserving federated approach for analyzing FAIR data in healthcare |
| Arefolov et al. | 2021 | Implementation of The FAIR Data Principles for Exploratory Biomarker Data from Clinical Trials |
| Ofili et al. | 2021 | The Research Centers in Minority Institutions (RCMI) Consortium: A Blueprint for Inclusive Excellence |
| Haendel et al. | 2021 | The National COVID Cohort Collaborative (N3C): rationale, design, infrastructure, and deployment |
| Kumar et al. | 2021 | Federated Learning Systems for Healthcare: Perspective and Recent Progress |
| Abuja et al. | 2021 | Public–Private Partnership in Biobanking: The Model of the BBMRI-ERIC Expert Centre |
| Schulman et al. | 2021 | The Finnish Biodiversity Information Facility as a best-practice model for biodiversity data infrastructures |
| Cooper et al. | 2021 | Perspective: The Power (Dynamics) of Open Data in Citizen Science |
| Øvrelid et al. | 2021 | TSD: A Research Platform for Sensitive Data |
| Hanisch et al. | 2021 | Research Data Framework (RDaF): Motivation, Development, and A Preliminary Framework Core |
| Read et al. | 2021 | Embracing the value of research data: introducing the JCHLA/JABSC Data Sharing Policy |
| Hanke et al. | 2021 | In defense of decentralized research data management |
| Zegers et al. | 2021 | Mind Your Data: Privacy and Legal Matters in eHealth |
| Groenen et al. | 2021 | The *de novo* FAIRification process of a registry for vascular anomalies |
| Delgado Mercè et al. | 2021 | Approaches to the integration of TRUST and FAIR principles |
| Jeliazkova et al. | 2021 | Towards FAIR nanosafety data |
| Demchenko et al. | 2021 | Future Scientific Data Infrastructure: Towards Platform Research Infrastructure as a Service (PRIaaS) |

another 5% discussed solutions for 'Business' purposes. The 'Social Science', 'Environmental', 'Astronomy' and 'Nanotechnology' domains only had one publication each, and 10% of papers did not belong to a specific Field of Research, but involved solutions related to a 'General' use of restricted data.

## Overview of data methods

In the following paragraphs, we report the results of the data methods encountered in the included papers. During the data analysis, we found that the data methods could reasonably be modelled along with the steps of what we could call the "data life cycle". The first step refers to the data collection process and includes methods such as the application of standards and requesting consent from the data subjects. Once the collection is completed, the second step refers to the processing of the data and includes, for example, methods related to the curation and validation and the creation of synthetic data. Then the data is published, through methods such as the selection of appropriate repositories and federated systems, or the application of an embargo on data release. Finally, the data is used, through methods such as the employment of access control systems and the selection of secure environments. After data usage, there might also be post-usage methods employed, for example, the acknowledgement of the data owners as well as archiving any secondary results. Within the data collection step, an important type of method deals with the aspect of metadata representation, for example, methods describing the licenses and usage terms applicable to the data, the versions available and the provenance. Other important aspects of restricted data are anonymization methods, which happen during the data processing step. Such methods include, for example, the de-identification, the minimization and the pseudonymization of the data. In the sections that follow, we describe in more detail each of the steps of the data life cycle and their related methods. Also, we propose a graphical representation of the methods in Fig. 2, and an overview of the results can be seen in Table 3.

### *Data collection*

The first step in any data-related activity is to collect the data. During this step, many methods are relevant to facilitate inter-disciplinary cooperation and data reuse. For example, methods that involve applying standards, common formats and best practices while collecting the data. We have found that 19 publications (47.5%) mentioned such methods, which we have collectively called "Data Standardization" methods. Other methods to improve the cooperation across disciplines are the ones related to making connections between the data and already available semantic vocabularies, such as the European Language Social Science Thesaurus (ELSST) (*Balkan et al., 2011*). Such methods usually require the data collector to research how concepts about the data, such as variables or descriptors, can be best mapped to semantic vocabularies. This process often necessitates some type of experience with Linked Data, as the exact connection and mapping are not always already available and there might be the need of creating custom links. These methods have been collectively defined as "Semantic Mapping", and they have been found in a total of 12 publications (30%). Moreover, during the data collection step,

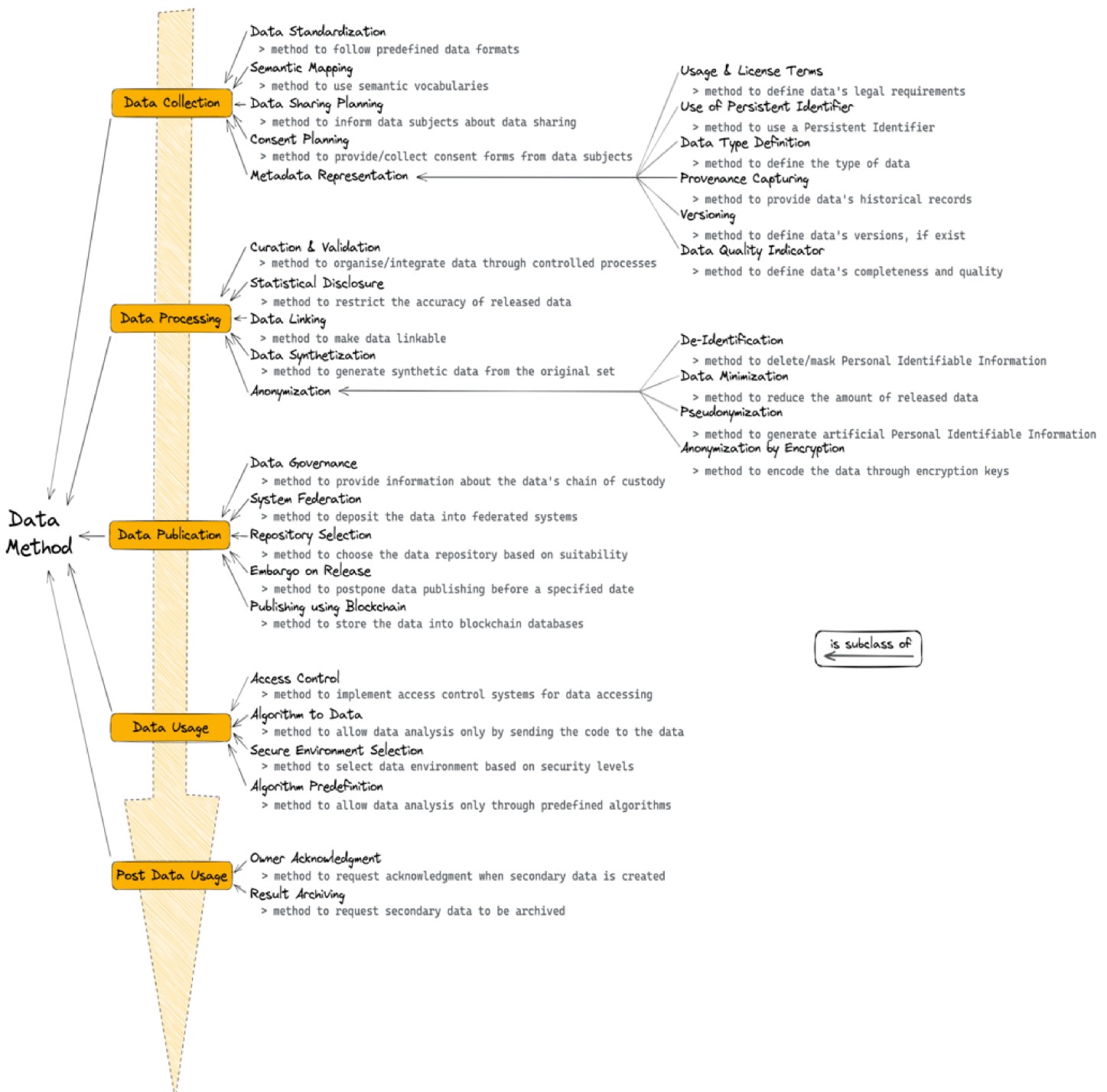

**Figure 2 Visual representations of the methods classes found during data analysis.** Below each method a short description can be found. Note the 'is subclass of' relations.

we also found methods about the request of the consent of collecting and sharing the data from the data subjects. Both types of methods can be applied to practically all types of data, but they possibly have more impact when the data collected contains Personal

**Table 3  Visual representation of the frequency of each method found in the included publications.**

| Authors | Year | TRL | Field of Research: Biomedical | General | Biodiversity | Business | Social Science | Humanity | Environmental | Astronomy | Nanotechnology | Data Collection: Data Standardization | Semantic Mapping | Data Sharing Planning | Consent Planning | Metadata Rep.: Usage & License Terms | Use of Persistent Identifier | Data Type Definition | Provenance Capturing | Versioning | Data Quality Indicator | Other Metadata Representations | Data Processing: Curation & Validation | Statistical Disclosure | Data Linking | Data Synthetization | Anonymization: De-Identification | Data Minimization | Pseudonymization | Anonymization by Encryption | Other Anonymizations | Data Publication: Data Governance | System Federation | Repository Selection | Embargo on Release | Publishing using Blockchain | Data Usage: Access Control | Algorithm to Data | Secure Environment Selection | Algorithm Predefinition | Post Usage: Owner Acknowledgement | Result Archiving | Methods Count (max = 31) | Classes Count (max = 7) |
|---|---|---|---|---|---|---|---|---|---|---|---|---|---|---|---|---|---|---|---|---|---|---|---|---|---|---|---|---|---|---|---|---|---|---|---|---|---|---|---|---|---|---|---|---|
| Dyke et al. | 2016 | 1 – 2 | X | | | | | | | | | | X | | X | | | | | | | X | X | | | | | | | | | | | | | | X | | | | | | 5 | 4 |
| Lakerveld et al. | 2017 | 5 – 6 | X | | | | | | | | | X | X | X | | X | | | | | | X | | | | | | | | | | | | | | | X | | | | | | 5 | 3 |
| Corpas et al. | 2018 | 1 – 2 | X | | | | | | | | | | | | X | X | X | X | | | X | | | | | | | | | | X | | X | | | X | | | | | | 6 | 3 |
| Bertocco et al. | 2018 | 3 – 4 | | | | | | | | X | | | | | | | | | | | | | | | | | | | | | | X | | | | | X | | | | | | 2 | 2 |
| Van Atteveldt et al. | 2019 | | | | | | X | | | | | X | | | | X | X | X | X | X | | X | X | | | | | | | | X | | | | | | X | X | X | X | | | 13 | 4 |
| Rockhold et al. | 2019 | 1 – 2 | X | | | | | | | | | | | X | | | | | | | | | | | | | | | | | | X | | | | | | | | | | | 2 | 2 |
| Demotes-Mainard et al. | 2019 | | X | | | | | | | | | | | X | | X | | | | | | | | | | | X | X | X | | X | X | | X | | | X | | | | | | 8 | 5 |
| Sun et al. | 2019 | 3 – 4 | X | | | | | | | | | | | | | | | | | | X | | | | | X | X | | | | | X | | | | | X | X | | | | | 5 | 3 |
| Dimper et al. | 2019 | 3 – 4 | X | | | | | | | | | | | X | | | | X | | | X | X | | | | | | | | | | X | | | | | X | | | | | | 5 | 4 |
| Becker et al. | 2019 | 5 – 6 | X | | | | | | | | | | | X | X | X | X | X | X | | | | | | | | | | | | | | | | | | X | | | | | | 5 | 3 |
| Lahti et al. | 2019 | 7 – 9 | | | X | | | | | | | | | X | | | | | | | | | X | | | | | | | | | | | | X | | X | | | | | 4 | 4 |
| Kleemola et al. | 2019 | 7 – 9 | | X | | | | | | | | X | | | X | | | X | | | X | X | | | | | | | | | | X | | | | | X | | | | | | 7 | 5 |
| Suver et al. | 2020 | | X | | | | | | | | | | | | | | | X | | | | | | | | X | | | | | | X | | | | | X | | | | | | 5 | 4 |
| Roche et al. | 2020 | | | | | | | X | | | | X | X | | | X | | | | | X | | | | | | | | X | | X | X | | X | | | X | | | | | | 11 | 5 |
| Paprica et al. | 2020 | 1 – 2 | X | | | | | | | | | X | X | X | X | | | | | | X | | | | | | | | | | X | | | | | X | X | | | | | 8 | 4 |
| Hoffmann et al. | 2020 | 1 – 2 | X | | | | | | | | | X | X | | | X | | | | | | X | | | | X | | | | | X | | | | | X | | | | | | 6 | 3 |
| Cullinan et al. | 2020 | 1 – 2 | X | | | | | | | | | | | | X | X | | | | | | | | | | X | X | | | | X | | | | | X | | | | | | 5 | 4 |
| Aarestrup et al. | 2020 | 1 – 2 | X | | | | | | | | | X | X | X | X | | | | | | X | | | | | | | | | | X | X | | | X | X | X | X | X | | X | | 13 | 5 |
| Nicholson et al. | 2020 | | X | | | | | | | | | X | X | | | | | | | | X | | X | X | | | | X | | | | X | | | | X | | | | | | 7 | 4 |
| Jaddoe et al. | 2020 | 3 – 4 | X | | | | | | | | | X | | X | | | | | | | X | | | | | | | | | | X | X | | | X | | | | | | 6 | 3 |
| Bader et al. | 2020 | | | | | X | | | | | | X | | | | X | | | | | | | | | | | | | | | X | X | | | X | | | | | | 6 | 4 |
| Choudhury et al. | 2020 | 5 – 6 | X | | | | | | | | | X | X | | | | | | | | | | | | | | | | | | | | | | | X | | | | | | 3 | 2 |
| Kephalopoulos et al. | 2020 | 7 – 9 | X | | | | | | | | | X | X | | | X | X | | | | X | X | X | | | | | | | | | | | | | X | | | | | | 8 | 4 |
| Beyan et al. | 2020 | 7 – 9 | X | | | | | | | | | | | | | | | | | | | | | | | | | | | | | | | | | X | | | | | | 1 | 1 |
| Zegers et al. | 2021 | | X | | | | | | | | | | X | | | | | | | | X | | | | | | | | | | X | | | | | X | X | X | | X | | 7 | 4 |
| Read et al. | 2021 | | | X | | | | | | | | | X | | | | | | | | | | | | X | | | | | | X | | X | | | | | X | | X | | 5 | 5 |
| Ovrelid et al. | 2021 | | | | X | | | | | | | | X | | | | | | | | | | | | | | | | | | X | | | | | | | | | X | | 3 | 3 |
| Demchenko et al. | 2021 | 1 – 2 | | | | | X | | | | | | X | | | | | | | | | | | | | | | | | | X | | | | | | | | | X | | 4 | 4 |
| Hanisch et al. | 2021 | | | X | | | | | | | | | X | | | | X | X | | X | X | | | | | | | | | | X | | | | | | X | | | X | X | 10 | 6 |
| Delgado Mercè et al. | 2021 | | X | | | | | | | | | | X | | | X | X | | X | X | | | X | | | | | | | | X | | | | | | X | | | | X | 11 | 6 |
| Cooper et al. | 2021 | | | | | X | | | | | | | X | X | | | | | | | | | | | | | | | | | X | | | | | | X | | | X | | 5 | 4 |
| Schulman et al. | 2021 | | | | X | | | | | | | X | | | | X | X | | | X | | | | | | | | | | X | | | | | | X | | | | | 5 | 3 |
| Ofili et al. | 2021 | 3 – 4 | X | | | | | | | | | X | | | | X | | | | | | | | | | | | | | | X | | | | | X | | | | | | 4 | 4 |
| Hanke et al. | 2021 | 3 – 4 | | X | | | | | | | | X | | | | X | | | | X | X | | | | | | | | | | X | | | | | X | | | | | | 6 | 4 |
| Groenen et al. | 2021 | | X | | | | | | | | | X | X | X | | X | X | | | X | X | | | | | | | | | | X | | | | | X | | | | | | 9 | 3 |
| Jeliazkova et al. | 2021 | | | | | | | | | X | | X | X | | | X | X | | | | | | X | X | | | | | | | X | | | | | X | | | | | | 9 | 5 |
| Haendel et al. | 2021 | 5 – 6 | X | | | | | | | | | X | X | | | | | | | | X | X | | | | X | X | X | X | | X | | | | | X | | | | | | 10 | 5 |
| Arefolov et al. | 2021 | | X | | | | | | | | | | X | X | X | X | | | | | | | | | | | | | | X | | | | X | | | X | | | | 7 | 4 |
| Kumar et al. | 2021 | 7 – 9 | X | | | | | | | | | | | | | | | | | | X | | | | | | | X | | | | | | X | | X | | | | | 3 | 3 |
| Abuja et al. | 2021 | 7 – 9 | X | | | | | | | | | | | | | | | | | | X | | X | | | | X | | | | | | X | | | | | X | | X | | 5 | 5 |
| **Instances Count (max = 40)** | | | 27 | 4 | 2 | 2 | 1 | 1 | 1 | 1 | 1 | 19 | 12 | 10 | 9 | 20 | 9 | 7 | 7 | 5 | 3 | 19 | 12 | 8 | 5 | 4 | 5 | 3 | 3 | 2 | 7 | 15 | 8 | 5 | 3 | 1 | 29 | 7 | 5 | 1 | 5 | 2 | | |

**Note:**
If a publication presented the method assigned to the column, then it would show an 'X' coloured cell. The colour of the cell is in correspondence to the Technology Readiness Level (TRL) assigned to the given publication: from lightest (light blue - TRL 1 & 2) to darkest (dark blue - TRL 7, 8 & 9). At the bottom of the figure, there is a row showing the number of articles each method has been found in, also colour graded from dark (many instances) to light grey (few instances). Overall, this table shows that there are wide variations in the frequency of the methods, but also that the vast majority of methods present TRL scores of five and above. We can also see that the coverage of the methods is rather broad and they are approximately evenly distributed among each class.

Identifiable Information (PII). For instance, when collecting patients' data it is important to clearly request the consent for collecting and sharing with each individual, as well as to define how and for what purposes the data can be shared. The methods that refer to

the planning and collection of consent forms from the data subjects are categorised together with the term "Consent Planning", and we found nine publications (22.5%) mentioning them. Moreover, methods referring to the request of consent from the data subjects about the sharing of the data are collectively called "Data Sharing Planning", and they were found in a total of 10 publications (25%).

### Metadata representation

Under the Data Collection Class, we also found methods related to the description of the data's top-level information, which have been categorised together as "Metadata Representation". For example, we found methods for the process of describing usage and license terms in the metadata, which has a key role in the reusability of secondary data as it outlines how access can be granted and under which conditions. In fact, a clear description of the usage and license terms in the metadata is essential for limiting and setting boundaries for secondary users. We found a total of 20 publications (50%) mentioning such methods, which we have called with the term "Usage and License Terms". If we turn now to the other methods under the Metadata Representation class, we found that nine publications (22.5%) mentioned the use of "Persistent Identifier" in the metadata. The use of a persistent identifier was expected to have a higher rate, as it is a key component for the Findability and Interoperability of data. A possible explanation of why this method was found in less than a quarter of the publications, is because often the data is released as a result of a publication, which is usually accompanied by an identifier. Nevertheless, the publication identifier (*e.g.*, DOI) relates to the publication itself and not the data it might contain, and in the instance of the data being reused, assigning a persistent identifier also the data could greatly improve the Findability and Interoperability of such resource. Moreover, we found other methods related to the definition of the type of data the metadata is describing. For example, by clearly stating if the metadata is describing survey, questionnaire or tabular data. These methods have been collected under the term "Data Type Definition", and we found them in seven publications (17.5%). In the same number of publications, we also found methods related to the description of the provenance of the data, which is an important aspect of making restricted data more FAIR. Data provenance methods have been collectively called "Provenance Capturing", and they include methods through which the origin of the data is documented. Also included in the Metadata Representation class, we found methods regarding the reporting of other versions of the data ("Versioning", found in five publications), and also methods describing the quality of the data ("Data Quality Indicator", found in three publications). The latter can indicate a variety of factors referring to the conditions of the data, such as its completeness, uniformity or if it is free of missing values and outliers. Lastly, we found 19 publications (47.5%) that mentioned the importance of having detailed metadata but did not specify any of its specific aspects in particular. Nevertheless, this result suggests that almost half of the included publications recognise the positive impact of having comprehensive metadata, even if no specific features were mentioned.

### Data processing

We will now present the results from the second step of the data life cycle, which involves the processing of the data after it has been collected. We found a variety of methods related to the processes of transforming, modifying and therefore processing the data before the other steps of the life cycle. Some of the methods found in the included publications involved practices to curate and validate the data before it can be published into databases and cloud services. Overall, such methods are aimed to improve the overall consistency and quality of the data, and they are related to the Quality Indicator method found in the Data Collection class, as it can either positively or negatively affect the quality of the data. We collectively named these methods "Curation and Validation", and they were found in 12 of the included publications (30%). We also found methods related to statistical techniques for limiting the accuracy or adding noise to the data to prevent the release of identifiable information. Such methods are potentially more applicable to numerical and tabular data, but they can indeed be applied also to questionnaire or survey data, by deciding for example to disclose only parts or modified versions of the original data. We have grouped these methods as the "Statistical Disclosure" methods, and they were found in eight publications (20%). The next set of methods found is related to the process of linking the data to other data sources already available in repositories, such as Google Dataset Search (https://datasetsearch.research. google.com), or to make the data suitable to be linkable by others. The "Data Linking" method, found in five publications (12.5%) is related to the Semantic Mapping under the Data Collection class, in the way that both methods refer to the process of creating links between the data and already available knowledge. However, in the Semantic Mapping case, links are aimed to be created between semantic vocabularies and data concepts instead of between data sources like in the case of Data Linking. We also found methods related to the creation of synthetic data from the original data to eliminate the possibility of identifiable or confidential information being exposed. We have collectively named these methods "Data Synthetization", and they were found in four publications (10%). It is now important to clearly define the relationship between the Data Synthetization method and the Anonymization subclass, and why this method has not been included in the subclass. Some readers may have expected the creation of synthetic data to be aligned with the concept of data anonymization, but in the context of this paper, we have made a distinction between the two. The Anonymization subclass refers to methods that are aimed to sanitize the data and make it free of personally identifiable and confidential information. Nevertheless, such processes are often applied to fragments or sections of the data and maintain the non-identifiable information intact. In contrast, with the Data Synthetization method, the data is used as a template for a completely new, and free of confidential information, set of data. Therefore, Data Synthetization and Anonymization are different in the way that, to achieve the removal of identifiable information, the first method requires completely new data to be generated, and the second one, instead, can be applied only to a section of the original data.

## Anonymization

Under the Data Processing Class, we also found methods related to different techniques for anonymizing the data, which represent the "Anonymization" methods. An important aspect, here, is to clearly describe the differences between the methods, as they can often be confused and misinterpreted. Some of the methods refer to the process of removing Personal Identifiable Information (PII) to eliminate the links between the data subjects and the data itself. These methods have been grouped under the "De-Identification" method, and they were found in five of the included publications (12.5%). Next, we found methods related to the reduction of released information, therefore minimizing the original data to a non-personal identifiable version. We also found other methods related to the process of replacing PII with artificially generated information, also called pseudonyms. These methods follow the same concept as the Data Synthetization ones, in the sense that artificial or synthetic information is created to avoid confidential data being exposed. The difference between the two methods is the fact that, while synthetization is applied to the whole data, pseudonymization is only applied to the personally identifiable information. The methods related to "Data Minimization" and "Pseudonymization" were only found in three publications each (7.5%). The last set of methods relates to the process of encrypting the whole or parts of the data to limit access to confidential information and PII. Central to this type of encryption method is that the encrypting key has to be kept secure from undesired use and unauthorised access. the "Anonymization by Encryption" methods were found in two publications (5%). Lastly, we found seven publications (17.5%) that mentioned the use of anonymization techniques to process restricted and confidential data but did not provide clear details regarding the specific type of anonymization used.

## Data publication

In the following section, we will present the results of the methods we found belonging to the third step of the data life cycle, which involves the publication of the collected data after it has been processed. Under the "Data Publication" step, we found methods related to the description of the data's chain of custody, also called "Data Governance" methods. Data governance aims to describe the standards by which the data is gathered, stored and processed, as well as to establish the responsibilities and authorities for its conservation. These methods, found in 15 publications (37.5%), are related to a variety of other methods found in different classes, such as the Usage and License Terms and Access Control methods, as they aim to improve the safeguarding of the data and to ensure its appropriate use. Next, we found methods related to the process of publishing the data into federated systems and allowing for the data to be combined with other resources, therefore improving both its Findability and Reusability. The "System Federation" method was found in eight publications (20%). Other methods that are relevant to the enhancement of FAIR, involve the decision of selecting the most appropriate repository or database to publish the data in. The "Repository Selection" method was only found in five publications (12.5%), and this low number could be explained by the fact that scientists do not find the selecting of the appropriate repository as a difficult task. Possibly, domain experts have a

clear understanding of the most used and most reliable repository, and therefore do not have to go through lengthy deliberation to agree on where the data can be published. We also found methods referring to the delaying or postponing of the publishing of the data, to minimise the effect of the data with respect to the time it was created. For example, by postponing the release of information by a couple of years, it is possible that identifiable information is not relevant anymore or that the data does not comport confidentiality issues any longer. Such methods have collectively been called "Embargo on Release" and they were found in three publications (7.5%). Lastly, we found only one publication referring to the process of making the data available through the adoption of a decentralised and distributed system using blockchain, to track and record data sharing and usage. The "Publishing using Blockchain" method is often a complex task, and it can require high technical skills and expertise, which could explain why this method was only found in one publication.

### Data usage

Once the data is published, it can also be used. The fourth step of the data life cycle represents the "Data Usage" step, and it includes a variety of methods involving the access and use of the data. For example, we found that most publications mentioned techniques for limiting access to restricted data to avoid undesired or unauthorised use. Such methods are also related to the Usage and License Terms method under the Metadata Representation class. In fact, the type of use that is allowed on the data can also influence the type of access requirements. For example, a specific dataset can be allowed to be used only for research purposes by university researchers, and this could be defined as the type of access requirements by only allowing access to the data to, for example, registered university researchers. This "Access Control" method was the most common among all the methods and all the life cycle steps, with a total of 29 publications (72.5%) mentioning it. Next, we found methods referring to the process of moving the analysis to where the data is stored. More specifically, in most common cases the data is accessed and stored or downloaded into personal machines or cloud systems, where then the data analysis is performed. Through the "Algorithm to Data" method, the data is never fully accessed by the user and cannot be downloaded. Instead, the user is only able to send their algorithms to the data to perform data analysis. This method, usually, does not require access control systems to be in place, as the analysis is most often allowed to return only aggregate results and, therefore, the concerns for the unintentional release of confidential and private information are limited. An example of the Algorithm to Data method is the Personal Health Train, a tool that allows the distributed analysis of health data while preserving privacy protection and ensuring a secure infrastructure (*Beyan et al., 2020*). The Algorithm to Data method was found in seven publications (17.5%). We also found methods involving the establishment or selection of safe infrastructure to allow for data usage. This "Secure Environment Selection" process, found in five publications (12.5%), can often comprise of a secure virtual or physical machine that limits the type of use granted, such as not allowing the download or the sharing of the data as well as limiting the information available for analysis. Lastly, we found one publication mentioning what we have called the

"Algorithm Predefinition" method, which refers to the process by which the data can only be analysed through a specific set of algorithms or statistical tests, that have been predefined *a priori* by the data owners.

### Post data usage

After the data is used and analysed, the last step of the data life cycle illustrates the methods describing what is required to be done for the "Post Data Usage". We found methods referring to the process of acknowledging the data owners by, for example, including information about the archive hosting the data or citing the original source. Next, we also found methods referring to the requirement from the data owners to archive the results from the analysis (also called secondary data) into the same repository as the original data. The "Owner Acknowledgement" method was found in five publications (12.5%) and the "Result Archiving" method was found in only two publications (5%).

## Technology readiness level (TRL)

A technology readiness level (TRL) was estimated for each publication based on the maturity of each system. As mentioned in the Results section, the TRLs were classified into four groups: TRL 1 & 2, TRL 3 & 4, TRL 5 & 6 and TRL 7, 8 & 9. The higher the TRL, the more mature the research infrastructures proposed in the publications are. By grouping the included papers by year, we have found that the ones published between 2019 and 2021 had the full array of TRLs. This means that at least one publication published each year had the lowest level (TRL 1 & 2) and at least one publication had the highest level (TRL 7 to 9). For the year 2016, we found only one paper with TRL 1 & 2, and for the subsequent year (2017) we also found only one paper with TRL 5 & 6. In 2018, instead, we found two papers, one with TRL 1 & 2 and the other one with TRL 3 & 4. We decided to focus on the analysis of the methods proposed by papers with the highest TRL levels, 7 to 9. This decision was made based on the assumption that if a method was employed in an infrastructure tested and implemented in the real world, such a method represented a reliable and mature way of dealing with restricted research data.

Under the class Data Collection, we found that 'Data Standardization', 'Metadata Representation', 'Semantic Mapping' and 'Data Sharing Planning' are methods belonging to at least one publication with TRL 7 to 9. This means that all subclasses of the category Data Collection displayed the highest level of maturity, except for the 'Consent Planning' method. Further, four out of six methods under the Metadata Representation class had a TRL 7 to 9, 'Usage and License Terms', 'Use of PID', 'Provenance' and 'Data Quality'. With regards to the Data Processing and Anonymization classes, only four out of nine subclasses belonged to paper with a high TRL. The subclasses are 'Curation & Validation', 'Statistical Disclosure', 'Data Minimization' and 'Encryption'. Under the Data Publication and the Data Usage classes, we found that all subclasses except for 'Blockchain' and 'Algorithms Pre-Definition' belonged to at least one publication with TRL 7 to 9. Lastly, for the Post Usage class, no publications displayed the highest TRL level. Table 3, summarise the findings of the data collection process.

## Data methods ontology

An OWL ontology was generated to describe the Data Methods subclass hierarchy. The Data Methods ontology was constructed as follows: Data Methods is an owl:Class, and all methods are rdfs:subClassOf Data Methods. Each class has a skos:prefLabel indicating the English name of that class, and a skos:definition property describing the meaning of that method.

## Data access

A full list of the included and excluded publications can be found on Zenodo (https://doi.org/10.5281/zenodo.6323515) and the OWL ontology describing the Data Methods can be found at the following link (www.w3id.org/odissei/ns/datamethods).

## DISCUSSION

The next section of this paper will focus on the discussion of the main findings, starting from the "Field of Research" results and moving on to the "Data Methods" results. We will then focus on the results from the "Metadata Representation" method class and, lastly, we will be discussing some limitations encountered in this research.

## Field of research

The results from the Field of Research analysis have shown that the Biomedical domain was the most common among the included publications. In fact, 67.5% of the papers included in the final set of this review belonged to such a field. This result suggests that the application of FAIR in the context of restricted data has a major impact in the Biomedical field, which is understandable considering the type of data that this field often requires, *e.g.*, patient records and genetic data. Even though such results suggest the importance of FAIR in the domain, it can also be seen as a source of bias in the analysis. We can hypothesise that the methods found in the review are the ones most applied in the Biomedical Field, and do not fully represent the wide range of domains where FAIR is currently been applied.

## Method classes

The present study was designed to determine ways restricted research data is used in the context of the FAIR principles. The first question in this study sought to determine the methods employed and/or suggested in the literature to help overcome the barrier of restricted data. We found that the most common solutions are 'Access Control', 'Usage and License Terms', 'Data Standardization' and 'Metadata Representation'. Each of these methods has at least one publication with the highest level of maturity, TRL 7 to 9, suggesting that they are not only widely employed, but also that the system employing them has been tested and implemented in the real world.

Interestingly, we realised that the majority of methods proposed require to be implemented before, or at the moment of, data collection and creation. For example, under the class Data Collection, both Consent Planning and Data Sharing Planning methods are required to be applied before data collection and agreed upon during the study design.

Informing data subjects about sharing conditions and asking for their consent is, of course, a core element of reusing restricted data, but they can't be introduced once the study has already started or been published. Moreover, other methods found require the data stakeholder to have a high level of technical knowledge and to be able to apply certain processing techniques to the data. For example, the application of methods under the Data Processing class such as Anonymization and Data Synthetization, assume technical proficiency. More specifically, methods of Anonymization (De-Identification, Data Minimization, Pseudonymization and Anonymization by Encryption) are often only applied by experts in computing and statistics. A similar conclusion can also be drawn to methods belonging to the Data Publication and Data Usage Classes. System Federation and Secure Environment Selection methods require a pre-existing knowledge and experience of such technologies, which is often only available to big organizations and experts in the field. Further, the archiving of data and making sure that it is stored in a safe environment often calls for considerable investments that might not necessarily be available to the stakeholders.

Therefore, we would like to draw attention to those methods that we believe are less challenging to implement, and can also be introduced at any point of the data life cycle: the metadata representation methods.

## Metadata representation

To help the discoverability and reusability of restricted secondary data, we should be focusing on techniques that can be applied and implemented after the data is collected and created, and that do not require a high level of technical expertise. This would allow non-technical stakeholders to be comfortable in publishing safe restricted data as well as being able to make data that is already available, FAIR. The creation of extensively descriptive and FAIR metadata is key to this process. Because the metadata represents the top-level information of the data it describes, it can be created, expanded and modified *a posteriori*, and does not intrinsically impose confidentiality concerns. The methods found in this review belonging to the metadata representation class are clearly related to the FAIR Principles, as they allow for better Findability, Accessibility, Interoperability and Reusability of the resource. In more detail, information about the Usage and License Terms could help researchers to understand exactly what actions are allowed on the data and how to request access, and provenance capturing could give important information about the data owners and stakeholders. Details about different available versions of the data (Versioning), as well as the Use of a Personal Unique Identifier, could help with Interoperability, by clearly stating the exact data used for analysis. Moreover, data type definition and data quality indicator methods could give insights to the researcher about the type of data included in the dataset as well as its quality.

Overall, each of these methods can be implemented after that data has been released but, of course, it is advisable to have an optimal metadata structure at the very stage of the data life cycle. Extensive and highly descriptive metadata information is a key component of making restricted data FAIR, as they can be designed not to contain any confidential information, but still benefit the research community.

## Limitations

Several limitations need to be noted regarding the present study. Despite the review offering some meaningful insights into technical solutions to overcome the barrier to researching restricted data, it has certain limitations in terms of the selection process as well as data analysis and extraction. A potential source of bias in this study lies in the fact that only one author was primarily responsible for the application of the inclusion/exclusion criteria, as well as for the data extraction and analysis. Although the author tried to assess each publication objectively and methodically following the criteria, it is possible that different results would have been generated if more authors were part of the evaluation. Moreover, the vast majority of papers were excluded based on the 4th eligibility criteria, which was to "clearly describe the proposal or application of the FAIR principles in the context of restricted data". This suggests that even though the papers mentioned the FAIR principles within their abstracts or full text, we could not find a clear application of the principles regarding restricted data. As a possible extension to this work, it would be interesting to contact the authors of the excluded papers and perform a survey to better understand and verify the intentions and limitations of the application of the principles in this context.

## CONCLUSION

The present study set out to provide the first systematic account of the relationship between Open Science, restricted data and the FAIR principles. The findings of this research provide insights into different ways restricted research data can be used, shared, stored and analysed, by respecting the privacy concerns in the reality of the Open Science world. With our results, we are providing an overview of the methods used when using restricted data in a FAIR manner, as well as a categorisation of such methods in both human and machine readable formats. The Data Methods framework and ontology we developed, can be used in the future to comply with the FAIR principles and provide information on how research on restricted data has been developed.

If the debate is to be moved forward, a better understanding of how the information resulting from this review can help in further achieving FAIR in restricted research data is needed. More research is required to develop a modelling strategy for improving the Findability, Accessibility, Interoperability and Reusability of restricted data. The FAIR Principles have been widely used in a variety of fields, and many guidelines and frameworks have been proposed, such as the "Top 10 FAIR Data and Software Things" (*Erdmann et al., 2019*), the "FAIR Metrics" (*Wilkinson et al., 2018*), the "FAIR Data Maturity Model" (*FAIR Data Maturity Model Working Group, 2020*) and the Internet of FAIR Data and Services (IFDS) (https://www.go-fair.org/resources/internet-fair-data-services/). Nevertheless, no FAIR framework has yet been proposed that directly addresses the issues concerning research with confidential and restricted access data. It would be interesting to assess how the information about the Data Methods found in this review can be introduced in the metadata of restricted data, and investigate whether available metadata models are suitable for such implementation. In fact, metadata has a key role in the development of FAIR workflows and, as discussed previously, we believe that extensive

metadata information is also key for the reuse of restricted access data in a FAIR manner. We conclude that with the present systematic review we are providing a framework to organise our knowledge about the methods employed in restricted data research, highlighting the importance of Metadata Representation and the FAIR Principles. We hope that our results can find practical applications both for stakeholders and researchers, and the methods found can be implemented in future projects.

### Funding

This project is funded by the Netherlands Organisation of Scientific Research (NWO), ODISSEI Roadmap project: 184.035.014. The funders had no role in study design, data collection and analysis, decision to publish, or preparation of the manuscript.

### Grant Disclosures

The following grant information was disclosed by the authors:
Netherlands Organisation of Scientific Research.
ODISSEI Roadmap Project: 184.035.014.

### Competing Interests

The authors declare that they have no competing interests.

### Author Contributions

- Margherita Martorana conceived and designed the experiments, performed the experiments, analyzed the data, performed the computation work, prepared figures and/or tables, authored or reviewed drafts of the article, and approved the final draft.
- Tobias Kuhn conceived and designed the experiments, analyzed the data, prepared figures and/or tables, authored or reviewed drafts of the article, and approved the final draft.
- Ronald Siebes conceived and designed the experiments, analyzed the data, authored or reviewed drafts of the article, and approved the final draft.
- Jacco van Ossenbruggen conceived and designed the experiments, authored or reviewed drafts of the article, and approved the final draft.

### Data Availability

A full list of the included and excluded publications is available at Zenodo: Margherita Martorana. (2022). ritamargherita/DataMethods: added bib files (v0.0.2). Zenodo. https://doi.org/10.5281/zenodo.6323515.

The ontology is also available at GitHub: https://raw.githubusercontent.com/ritamargherita/DataMethods/main/DataMethods.ttl.

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
