# Peer review of "Aligning restricted access data with FAIR: a systematic review"

_PeerJ Computer Science, doi:10.7717/peerj-cs.1038_

## Round 0.1 · original submission · Minor Revisions

The manuscript could be re-submitted after addressing all the concerns raised by the reviewers.

Reviewer 1 ·

Basic reporting

1. To date, it remains unclear how sensitive data should be managed, accessed, and analysed (Cox et al., 2016).
- 6-year-old statement is given to express today’s situation

2. Is there any solid ground behind the selection of Eligibility criteria?
3. Condition four of the eligibility criteria is not present in a vast number of articles?

4. The corpus of publications resulting from the query was exported to Rayyan
- Why this specific tool is being used?
- This tool was presented in 2016. Do we have any latest tool available in this context?
- Comparative analysis is required for the selection of such tool from the list of available tools
- It is seemed that the proposed work is heavily depending on this tool. What if this tool is removed / replaced with some other tool?
- Which specific version of the tool is used?

5. In the context of this research, the cycle has been slightly modified to better suit our results and to bring more awareness to also the stages when the data is processed, as well as the steps required after the data is used.
- If there is no other reason of modifying data lifecycle management then it seems a bit biased to modify the process.
- Secondly, what alteration has been made?

6. TRL levels details are missing.
7. Line 244 to 251 can be better represented in a tabular form.
8. What is OWL ontology and why it is being used here?
9. An RDF ontology was generated to describe the Data Methods subclass hierarchy.
- Is there any alternate to the above representation?
- Why do we need this step?

Is the review of broad and cross-disciplinary interest and within the scope of the journal?
- Yes
Has the field been reviewed recently? If so, is there a good reason for this review (different point of view, accessible to a different audience, etc.)?
- No, the authors claim is that this is the first study of this type.
Does the Introduction introduce the subject and make it clear who the audience is/what the motivation is?
- Yes, up to some extent.

Experimental design

Methods described with sufficient detail & information to replicate.
- No, with the provided details it will be difficult to reduce the result because it is working on the result of google scholar. At different time, google scholar will return different papers. This will affect the result.

Is the Survey Methodology consistent with a comprehensive, unbiased coverage of the subject? If not, what is missing?

Are sources cited? Quoted or paraphrased as appropriate?
- Yes
Is the review organized logically into coherent paragraphs/subsections?
- Yes

Validity of the findings

Is there a well-developed and supported argument that meets the goals set out in the Introduction?
- In this context, goals are not being set out in the introduction.

Does the Conclusion identify unresolved questions / gaps / future directions?
- There is a limitation heading found in the article in this regard.

Overall, this article is divided into two parts:
- Domain / problem introduction
- Proposed Solution

First part is being presented very well. Covering subject area well e.g., introduction, problem statement, background etc. but when it comes to problem solution then some weak technical stuff is being observed e.g., Methods, eligibility criteria etc. The Results section contains nothing but the detailed explanation of the selected dataset. Similarly, the outcome of the paper is heavily depended on the selected dataset. E.g., different input articles will have different results.

Reviewer 2 ·

Basic reporting

The paper analyzes various practices being followed in the research community to meet the data sharing proposed by FAIR principles. The manuscript provides a detailed survey of key mechanisms involved in applying FAIR principles on restricted research data.

Experimental design

The identified research questions look appropriate however the utility of RQ2 and RQ3 should be further elaborated, i.e., how research community may take advantage of proposed categorization and whether the mature practices can be converted into some kind of standards or standard operation procedures or not?

The records show that majority of the papers are unable to meet the criteria set by the submitted manuscript and 40 articles are selected for investigation of research questions. Theses number suggests that research community is either not aware of FAIR or the principles are not believed to improve the quality of research. A survey may be used to contact the authors of the excluded papers to verify the intention of no following FAIR principles.

It is suggested to target papers that have received awards in conferences or journals in the study as most of the conferences consider reproducibility as key condition for such awards. Also it can be found out whether compliance of FAIR principles is a requirement for the such awards or not?

Recently a few journals have appeared that particularity target publication of datasets (Data in Brief). Authors may consider including journal or conference based criteria to evaluate further the research questions.

Validity of the findings

A comprehensive discussion has been provided based on the collected data along with limitations, however authors apparently have relied on the reader for developing the inferences related to the research questions. It is suggested to provide crisp response to the research questions particularly RQ2 and RQ3.

Authors have discussed the aspect of anonymization of data however it is suggested to investigate the aspect keeping finer granularity in mind, i.e., what kind of anonymization principles should be or already part of FAIR?

Table 2 provides fields of research categories and no example can be found that belongs to multiple fields. As the manuscript is submitted in a computer science journal, it is therefore suggested to make effort to find a few paper that belong to either core computer science or its constituents field, i.e. communication networks, social network analysis, software project management etc that usually include research works involving sharing of data.

---

## Round 0.2 · accepted · Accept

Congratulations, the revised version of the manuscript is satisfactory and it is recommended for publication.